# TASK-SIMILARITY AWARE META-LEARNING THROUGH NONPARAMETRIC KERNEL REGRESSION

## ABSTRACT

This paper investigates the use of nonparametric kernel-regression to obtain a task-similarity aware meta-learning algorithm. Our hypothesis is that the use of task-similarity helps meta-learning when the available tasks are limited and may contain outlier/ dissimilar tasks. While existing meta-learning approaches implicitly assume the tasks as being similar, it is generally unclear how this task-similarity could be quantified and used in the learning. As a result, most popular meta-learning approaches do not actively use the similarity/dissimilarity between the tasks, but rely on availability of huge number of tasks for their working. Our contribution is a novel framework for meta-learning that explicitly uses task-similarity in the form of kernels and an associated meta-learning algorithm. We model the task-specific parameters to belong to a reproducing kernel Hilbert space where the kernel function captures the similarity across tasks. The proposed algorithm iteratively learns a meta-parameter which is used to assign a task-specific descriptor for every task. The task descriptors are then used to quantify the task-similarity through the kernel function. We show how our approach conceptually generalizes the popular meta-learning approaches of model-agnostic meta-learning (MAML) and Meta-stochastic gradient descent (Meta-SGD) approaches. Numerical experiments with regression and classification tasks show that our algorithm outperforms these approaches when the number of tasks is limited, even in the presence of outlier or dissimilar tasks. This supports our hypothesis that task-similarity helps improve the meta-learning performance in task-limited and adverse settings.

## 1 INTRODUCTION

Meta-learning seeks to abstract a general learning rule applicable to a class of different learning problems or tasks, given the knowledge of a set of training tasks from the class (Finn & Levine, 2018; Denevi et al., 2018; Hospedales et al., 2020; Grant et al., 2018; Yoon et al., 2018). The setting is such that the data available for solving each task is often severely limited, resulting in a poor performance when the tasks are solved independently from each other. This also sets meta-learning apart from the transfer learning paradigm where the focus is to transfer a well-trained network from existing domain to another (Pan & Yang, 2010). While existing meta-learning approaches implicitly assume the tasks as being similar, it is generally unclear how this task-similarity could be quantified and used in the learning. As a result, most popular meta-learning approaches do not actively use the similarity/dissimilarity between the tasks, but rely on availability of huge number of tasks for their working. In many practical applications, the number of tasks could be limited and the tasks may not always be very similar. There might even be 'outlier tasks' or 'out-of-the-distribution tasks' that are less similar or dissimilar from the rest of the tasks. Our conjecture is that *the explicit incorporation or awareness of task-similarity helps improve meta-learning performance in such task-limited and adverse settings*.

The goal of this paper is to test this hypothesis by developing a task-similarity aware meta-learning algorithm using nonparametric kernel regression. Specifically, our contribution is a novel meta-learning algorithm called the *Task-similarity Aware Nonparametric Meta-Learning* (TANML) that:

- Explicitly *employs similarity* across the tasks to fast adapt the meta-information to a given task, by using nonparametric kernel regression.

- Models the parameters of a task as belonging to a reproducing kernel Hilbert space (RKHS), obtained by viewing the popular meta-learning of MAML and Meta-SGD approaches through the lens of linear/kernel regression.
- Uses *task-descriptors* through a kernel function to quantify task-similarity/dissimilarity.
- Offers a *general framework* for incorporating task-similarity in the meta-learning process. Though we pursued the algorithm with a specific choice of the task-descriptors, the proposed RKHS task-similarity aware framework can be extended to other formulations.

We wish to emphasize that our goal is *not* to propose another meta-learning algorithm that outperforms the state-of-the-art, but rather to investigate if task-similarity can be explicitly incorporated and used to advantage in a meaningful manner. We show how this is achieved as the consequence of viewing the popular MAML and Meta-SGD formulations through the lens of nonparametric kernel regression. In order to keep the comparison fair on an apple-to-apple level, we compare the performance of TANML with that of MAML and Meta-SGD algorithms.

## 1.1 MATHEMATICAL OVERVIEW OF THE PROPOSED TASK-SIMILARITY AWARE FRAMEWORK

Given pairs of data $(x_k, y_k) \in \mathbb{R}^{n_x} \times \mathbb{R}^{n_y}$ where $k \in \{1, 2, \cdots, K\}$ generated by an unknown data source, we are interested in learning a predictor $\mathbb{R}^{n_x} \times \mathbb{R}^D \ni (x, \boldsymbol{\theta}) \mapsto f(x, \boldsymbol{\theta}) \in \mathbb{R}^{n_y}$ from the given data. For example, $f(x, \boldsymbol{\theta})$ could be a function defined by an artificial neural network (ANN). We collect pairs of data in $\mathcal{X} = (x_1, x_2, \cdots, x_K)$ and $\mathcal{Y} = (y_1, y_2, \cdots, y_K)$ and define the loss function $\mathbb{R}^{Kn_x} \times \mathbb{R}^{Kn_y} \times \mathbb{R}^D \ni (\mathcal{X}, \mathcal{Y}, \boldsymbol{\theta}) \mapsto \mathcal{L}(\mathcal{X}, \mathcal{Y}, \boldsymbol{\theta}) \in \mathbb{R}$ which we then minimize with respect to $\boldsymbol{\theta}$. This constitutes the training of the predictor. In the case of a ANN, $\mathcal{L}(\mathcal{X}, \mathcal{Y}, \boldsymbol{\theta}) \in \mathbb{R}$ could be the mean-square loss or the cross-entropy function. The data $\mathcal{X}, \mathcal{Y}$ used for training is referred to as the training data. Let $\hat{\boldsymbol{\theta}}$ denote the optimal value of $\boldsymbol{\theta}$ obtained from training. Given a new $x \in \mathbb{R}^{n_x}$, we use $\hat{y} = f(x, \hat{\boldsymbol{\theta}})$ to predict $y$. The goodness of $\hat{\boldsymbol{\theta}}$ is evaluated using $y - \hat{y}$ on a sequence of pairs of new data called the test data $\bar{\mathcal{X}}, \bar{\mathcal{Y}}$, defined similarly as $\mathcal{X}$ and $\mathcal{Y}$, but with $\bar{K}$ number of data pairs. The training of the predictor for the given data source is defined as a task.

Now, consider that we are interested in carrying out several such tasks for data coming from different but similar sources. Let $\mathcal{X}_i, \mathcal{Y}_i, \bar{\mathcal{X}}_i, \bar{\mathcal{Y}}_i, i = 1, \cdots, T_{tr}$ denote the data from $T_{tr}$ different data-sources, and defined similarly as $\mathcal{X}, \mathcal{Y}, \bar{\mathcal{X}}, \bar{\mathcal{Y}}$ above. We refer to the training of the predictor for data $\mathcal{X}_i, \mathcal{Y}_i, \bar{\mathcal{X}}_i, \bar{\mathcal{Y}}_i$ as the $i$th training task, and $\boldsymbol{\theta}_i$ is referred to as the parameter for the task. Meta-learning captures similarities across the tasks by learning a common $\hat{\boldsymbol{\theta}}$ (which we denote by $\boldsymbol{\theta}_0$) from the data of these $T_{tr}$ tasks (called the meta-training data), such that $\boldsymbol{\theta}_0$ can be quickly adapted to train a predictor for data from new and different but similar data-sources. Depending on how $\boldsymbol{\theta}$ is obtained from $\boldsymbol{\theta}_0$, various meta-learning algorithms exist (Denevi et al., 2018; Finn & Levine, 2018; Allen et al., 2019). The performance of the meta-learning algorithm is evaluated on previously unseen data from several other similar sources $\mathcal{X}_i^v, \mathcal{Y}_i^v, \bar{\mathcal{X}}_i^v, \bar{\mathcal{Y}}_i^v, i = 1, \cdots, T_v$ (called the meta-test data) defined similarly to $\mathcal{X}, \mathcal{Y}, \bar{\mathcal{X}}, \bar{\mathcal{Y}}$ − this constitutes the meta-test phase. The training of the predictor for test data $\mathcal{X}_i^v, \mathcal{Y}_i^v, \bar{\mathcal{X}}^v{}_i, \bar{\mathcal{Y}}^v{}_i$ is referred to as the $i$th test task, $\boldsymbol{\theta}_i^v$ denotes the parameter for the $i$th test task. In the existing meta-learning approaches, both $\boldsymbol{\theta}_i$ and $\boldsymbol{\theta}_i^v$ are obtained by adapting $\boldsymbol{\theta}_0$ using the gradient of $\mathcal{L}(\mathcal{X}_i, \mathcal{Y}_i, \boldsymbol{\theta})$ and $\mathcal{L}(\mathcal{X}_i^v, \mathcal{Y}_i^v, \boldsymbol{\theta})$, respectively, evaluated at $\boldsymbol{\theta}_0$.

In our work, we propose a meta-learning algorithm where $\boldsymbol{\theta}_i$ explicitly uses a similarity between the $i$th training task and all the training tasks. Similarly, the parameters $\boldsymbol{\theta}_i^v$ for the test tasks also use explicitly a similarity between the $i$th test task and all the training tasks. As specified later, we define this *task-similarity* between two tasks through kernel regression, and our algorithm learns the kernel regression coefficients $\boldsymbol{\Psi}$ as meta-parameters in addition to $\boldsymbol{\theta}_0$.

**A motivating example** Let us now consider a specific loss function given by $\mathcal{L}(\mathcal{X}, \mathcal{Y}, \boldsymbol{\theta}) = \sum_{k=1}^{K} \|y_k - f(x_k, \boldsymbol{\theta})\|_2^2$. Training for tasks independently with limited training data will typically result in a predictor that overfits to $\mathcal{X}, \mathcal{Y}$, and generalizes poorly to $\bar{\mathcal{X}}, \bar{\mathcal{Y}}$. MAML-type meta-learning approaches (Finn et al., 2017) solve this by inferring the information across tasks in the form of a good initialization $\boldsymbol{\theta}_0$− specialized/adapted to a new task using the *adaptation function* $\mathbb{R}^D \times \mathbb{R}^{Kn_x} \times \mathbb{R}^{Kn_y} \ni (\boldsymbol{\theta}_0, \mathcal{X}, \mathcal{Y}) \mapsto g_{\text{MAML}}(\boldsymbol{\theta}_0, \mathcal{X}, \mathcal{Y}) \in \mathbb{R}^D$ defined as:

$$g_{\text{MAML}}(\boldsymbol{\theta}_0, \mathcal{X}, \mathcal{Y}) \triangleq \boldsymbol{\theta}_0 - \alpha \nabla_{\boldsymbol{\theta}_0} \mathcal{L}(\mathcal{X}, \mathcal{Y}, \boldsymbol{\theta}_0)$$

The parameters for the training and test tasks as obtained through adaptation of $\boldsymbol{\theta}_0$ as

$$\boldsymbol{\theta}_i = g_{\text{MAML}}(\boldsymbol{\theta}_0, \mathcal{X}_i, \mathcal{Y}_i), \ i = 1, \cdots, T_{tr}, \ \text{and} \quad \boldsymbol{\theta}_i^v = g_{\text{MAML}}(\boldsymbol{\theta}_0, \mathcal{X}_i^v, \mathcal{Y}_i^v), \ i = 1, \cdots, T_v.$$

The meta-parameter $\boldsymbol{\theta}_0$ is learnt by iteratively taking a gradient descent with respect to the test loss on the training tasks given by $\sum_{i=1}^{T_{tr}} \mathcal{L}(\bar{\mathcal{X}}_i, \bar{\mathcal{Y}}_i, g_{\text{MAML}}(\boldsymbol{\theta}_0, \mathcal{X}_i, \mathcal{Y}_i))$. The parameters for a task are obtained directly from $\boldsymbol{\theta}_0$ and does not make use of any information from the other training tasks. As a result, the common $\boldsymbol{\theta}_0$ learnt during the meta-training treats all tasks equally − the algorithm implicitly assumes similarity of all tasks, but is not able to discern or quantify the degree of similarity or dissimilarity among the tasks. In contrast, our algorithm involves an adaptation function $g_{\text{TANML}}$ (to be defined later) that explicitly uses a notion of similarity between the tasks to predict parameters for a task. As a result, we expect that our approach helps train predictors even when the data-sources that are not very similar to each other. In our numerical experiments in Section 4, we see that this is indeed the case the sinusoidal function as the data source.

## 1.2 RELATED WORK

The structural characterization of tasks and use of task-dependent knowledge has gained interest in meta-learning recently. Edwards & Storkey (2017) proposed a variational autoencoder based approach to generate task/dataset statistics used to measure similarity. Ruder & Plank (2017) considered domain similarity and diversity measures in the context of transfer learning (Ruder & Plank, 2017). The study of how task properties affect the catastrophic forgetting in continual learning was pursued by Nguyen et al. (2019). Lee et al. (2020) proposed a task-adaptive meta-learning approach for classification that adaptively balances meta-learning and task-specific learning differently for every task and class. Bayesian approaches have been proposed to capture the similarity across tasks in the form of task hyperpriors (Yoon et al., 2018; Finn et al., 2018; Grant et al., 2018; Rothfuss et al., 2020). Task-similarity defined through effective sample size has been used to develop a new off-policy algorithm for meta-reinforcement learning (Fakoor et al., 2020). It was shown by Oreshkin et al. (2018) that the performance few-shot learning shows significant improvements with the use of task-dependent metrics. While the use of kernels or similarity metrics is not new in meta-learning, they are typically seen in the context of defining relations between the classes or samples within a given task (Qiao et al., 2018; Rusu et al., 2019; Vinyals et al., 2016; Snell et al., 2017; Oreshkin et al., 2018; Fortuin & Rätsch, 2019; Goo & Niekum, 2020). Qiao et al. (2018) use similarity metrics in the activation space to predict parameters for novel categories in few-shot learning with zero-training. Information-theoretic ideas have also been used in the study of the topology and the geometry of task spaces by Nguyen et al. (2019); Achille et al. (2018). Achille et al. (2019) construct vector representations for tasks using partially trained probe networks, based on which task-similarity metrics are developed. Task descriptors have been of interest specially in vision related tasks in the context of transfer learning (Zamir et al., 2018; Achille et al., 2019; Tran et al., 2019). Recently, neural tangent kernels were been proposed for asymptotic analysis of meta-learning for infinitely wide neural networks by considering gradient based kernels across tasks by Wang et al. (2020).

The work of Wang et al. (2020) is the closest in spirit to our work in that they consider kernels across meta-learning tasks. However, the premise of their work is very different from ours. The aim of their work is to show how global convergence behaviour of popular MAML type task-specific adaptation can be assymptotically described using specific kernels, when every task involves training deep neural networks of asymptotically infinite or very large widths. Our premise on the other hand is entirely different − we consider a task-specific adaptation that actively employs similarity of tasks in the form of valid kernel functions, in order to improve meta-learning performance. Our work does not make assumptions on the nature of the kernel, the structure of the learner, or its dimensions.

## 2 REVIEW OF MAML AND META-SGD

To facilitate our analysis, we first review MAML and Meta-SGD approaches and highlight the relevant aspects necessary for our discussion. We shall then show how these approaches lead to the definition of a generalized meta-SGD and consequently, to our TANML approach.

### 2.1 META AGNOSTIC META-LEARNING

Model-agnostic meta-learning proceeds in two stages iteratively. As discussed in the motivating example, the parameter $\boldsymbol{\theta}_i$ for the $i$th training task $\mathcal{X}_i, \mathcal{Y}_i, \bar{\mathcal{X}}_i, \bar{\mathcal{Y}}_i, \ i = 1, \cdots, T_{tr}$ is obtained by

applying the adaptation function $g_{\mathrm{MAML}}$ to $\boldsymbol{\theta}_0$ as $\boldsymbol{\theta}_i = g_{\mathrm{MAML}}(\boldsymbol{\theta}_0, \mathcal{X}_i, \mathcal{Y}_i)$. This is called the inner update. Once $\boldsymbol{\theta}_i$ is obtained for all the training tasks, $\boldsymbol{\theta}_0$ is then updated by running one gradient descent step on the total test-loss given by $\sum_{i=1}^{T_{tr}} \mathcal{L}(\bar{\mathcal{X}}_i, \bar{\mathcal{Y}}_i, g_{\mathrm{MAML}}(\boldsymbol{\theta}_0, \mathcal{X}_i, \mathcal{Y}_i))$. This is called the outer update. Each outer update involves $T_{tr}$ inner updates corresponding to the $T_{tr}$ training tasks. The outer updates are run for $N_{iter}$ iterations. This constitutes the meta-training phase of MAML, described in Algorithm 1. Once the meta-training phase is complete and $\boldsymbol{\theta}_0$ is learnt, the parameters for a new test task are obtained by applying the inner update on the training data of the test task. We note here that MAML described in Algorithm 1 is the first-order MAML (Finn et al., 2018), as opposed to the general MAML where the inner update may contain several gradient descent steps. We note that when we talk of MAML in this paper, we always refer to the first-order MAML. A schematic of MAML is presented in Figure 1.

---

**Algorithm 1:** Model agnostic meta-learning (MAML)

---

Initialize $\boldsymbol{\theta}_0$
**for** $N_{iter}$ iterations **do**
    **for** $i = 1, \cdots, T_{tr}$ **do**
        | $g_{\mathrm{MAML}}(\boldsymbol{\theta}_0, \mathcal{X}_i, \mathcal{Y}_i) = \boldsymbol{\theta}_0 - \alpha \nabla_{\boldsymbol{\theta}_0} \mathcal{L}(\mathcal{X}_i, \mathcal{Y}_i, \boldsymbol{\theta}_0)$    [Inner update]
    **end**
    $\boldsymbol{\theta}_0 = \boldsymbol{\theta}_0 - \beta \nabla_{\boldsymbol{\theta}_0} \sum_{i=1}^{T_{tr}} \mathcal{L}(\bar{\mathcal{X}}_i, \bar{\mathcal{Y}}_i, g_{\mathrm{MAML}}(\boldsymbol{\theta}_0, \mathcal{X}_i, \mathcal{Y}_i))$    [Outer update]
**end**

---

## 2.2 Meta-Stochastic Gradient Descent (Meta-SGD)

Meta stochastic gradient descent (Meta-SGD) is a variant of MAML that learns the component-wise step sizes for the inner update jointly with $\boldsymbol{\theta}_0$. Let $\boldsymbol{\alpha}$ denote the vector of step-sizes for the different components of $\boldsymbol{\theta}$. As with MAML, the meta-training phase of Meta-SGD also involves an inner and an outer update. The outer update computes the values of $\boldsymbol{\theta}_0$ and $\boldsymbol{\alpha}$; the inner update computes the parameter values $\boldsymbol{\theta}_i$ using the adaptation function $\mathbb{R}^D \times \mathbb{R}^D \times \mathbb{R}^{Kn_x} \times \mathbb{R}^{Kn_y} \ni (\boldsymbol{\theta}_0, \boldsymbol{\alpha}, \mathcal{X}_i, \mathcal{Y}_i) \mapsto g_{\mathrm{MSGD}}(\boldsymbol{\theta}_0, \boldsymbol{\alpha}, \mathcal{X}_i, \mathcal{Y}_i) \in \mathbb{R}^D$ defined as

$$g_{\mathrm{MSGD}}(\boldsymbol{\theta}_0, \boldsymbol{\alpha}, \mathcal{X}_i, \mathcal{Y}_i) \triangleq \boldsymbol{\theta}_0 - \boldsymbol{\alpha} \cdot \nabla_{\boldsymbol{\theta}_0} \mathcal{L}(\mathcal{X}_i, \mathcal{Y}_i, \boldsymbol{\theta}_0),$$

where $\cdot$ operator denotes the point-wise vector product. The outer update is run for $N_{iter}$ iterations. The meta-training phase for Meta-SGD is described in Algorithm 2:

---

**Algorithm 2:** Meta-stochastic gradient descent

---

Initialize $[\boldsymbol{\theta}_0, \boldsymbol{\alpha}]$
**for** $N_{iter}$ iterations **do**
    **for** $i = 1, \cdots, T_{tr}$ **do**
        | $g_{\mathrm{MSGD}}(\boldsymbol{\theta}_0, \boldsymbol{\alpha}, \mathcal{X}_i, \mathcal{Y}_i) = \boldsymbol{\theta}_0 - \boldsymbol{\alpha} \cdot \nabla_{\boldsymbol{\theta}_0} \mathcal{L}(\mathcal{X}_i, \mathcal{Y}_i, \boldsymbol{\theta}_0)$    [Inner update]
    **end**
    $[\boldsymbol{\theta}_0, \boldsymbol{\alpha}] = [\boldsymbol{\theta}_0, \boldsymbol{\alpha}] - \beta \nabla_{[\boldsymbol{\theta}_0, \boldsymbol{\alpha}]} \sum_{i=1}^{T_{tr}} \mathcal{L}(\bar{\mathcal{X}}_i, \bar{\mathcal{Y}}_i, g_{\mathrm{MSGD}}(\boldsymbol{\theta}_0, \boldsymbol{\alpha}, \mathcal{X}_i, \mathcal{Y}_i))$    [Outer update]
**end**

---

The predictor for the $i$th test task is then trained by applying the inner update on $\mathcal{X}_i^v, \mathcal{Y}_i^v$, using the values of $\boldsymbol{\theta}_0$ and $\boldsymbol{\alpha}$ obtained in the meta-training phase.

We notice that the inner update is expressible as $g_{\mathrm{MSGD}}(\boldsymbol{\theta}_0, \boldsymbol{\alpha}, \mathcal{X}_i, \mathcal{Y}_i) = \mathbf{W}^\top \mathbf{z}_i(\boldsymbol{\theta}_0)$ where

$$\mathbf{W} \triangleq [\mathbf{I}, -\mathrm{diag}(\boldsymbol{\alpha})] \quad \text{and} \quad \mathbf{z}_i(\boldsymbol{\theta}_0) \triangleq \left[ \boldsymbol{\theta}_0^\top \ \nabla_{\boldsymbol{\theta}_0} \mathcal{L}(\mathcal{X}_i, \mathcal{Y}_i, \boldsymbol{\theta}_0)^\top \right]^\top. \tag{1}$$

The matrix $\mathbf{W}^\top$ denotes the transpose of $\mathbf{W}$, $\mathbf{I}$ denotes the identity matrix, and $\mathrm{diag}(\boldsymbol{\alpha})$ denotes the diagonal matrix whose diagonal is equal to the vector $\boldsymbol{\alpha}$. We refer to $\mathbf{z}_i(\boldsymbol{\theta}_0)$ as the *task descriptor* of the $i$th training task. Thus, $g_{\mathrm{MSGD}}(\boldsymbol{\theta}_0, \boldsymbol{\alpha}, \mathcal{X}_i, \mathcal{Y}_i)$ takes the form of a linear predictor for $\boldsymbol{\theta}_i$ with $\mathbf{z}_i(\boldsymbol{\theta}_0)$ as the input and regression coefficients $\mathbf{W}$. The adaptation $g_{\mathrm{MSGD}}(\boldsymbol{\theta}_0, \boldsymbol{\alpha}, \mathcal{X}_i, \mathcal{Y}_i)$ can be generalized to the case of $\mathbf{W}$ being a full matrix that is to be learnt from the training tasks. This generalization results in the adaptation function $\mathbb{R}^D \times \mathbb{R}^{D \times 2D} \times \mathbb{R}^{Kn_x} \times \mathbb{R}^{Kn_y} \ni (\boldsymbol{\theta}_0, \mathbf{W}, \mathcal{X}, \mathcal{Y}) \mapsto g_{\mathrm{GMSGD}}(\boldsymbol{\theta}_0, \mathbf{W}, \mathcal{X}, \mathcal{Y}) \in \mathbb{R}^D$ given by

$$g_{\mathrm{GMSGD}}(\boldsymbol{\theta}_0, \mathbf{W}, \mathcal{X}_i, \mathcal{Y}_i) = \mathbf{W}^\top \mathbf{z}_i(\boldsymbol{\theta}_0) = \mathbf{W}_1^\top \boldsymbol{\theta}_0 + \mathbf{W}_2^\top \nabla_{\boldsymbol{\theta}_0} \mathcal{L}(\mathcal{X}_i, \mathcal{Y}_i, \boldsymbol{\theta}_0)$$

where $\mathbf{W}_1, \mathbf{W}_2 \in \mathbb{R}^{D \times T}$ are the submatrices of $\mathbf{W}$ such that $\mathbf{W} = [\mathbf{W}_1 \, \mathbf{W}_2]$. Expressed in this manner, we notice how $g_{\text{GMSGD}}$ performs a parameter update similar to a second-order gradient update with $\mathbf{W}_2$ taking a role similar to the Hessian matrix. We refer to the resulting meta-learning algorithm as the *Generalized Meta-SGD* described in Algorithm 3. The second term $\Omega(\mathbf{W})$ in the outer loop cost function is a regularization that ensures $\mathbf{W}$ is bounded and avoids overfitting. On setting $\mu = 0$ and using $\mathbf{W}$ as defined in equation 1, the Generalized Meta-SGD reduces to the Meta-SGD. The Generalized Meta-SGD is thus a more general form of MAML arrived at by viewing MAML/Meta-SGD as a linear regression.

---

**Algorithm 3:** Generalized Meta-SGD

---

Initialize $[\boldsymbol{\theta}_0, \mathbf{W} \in \mathbb{R}^{2D \times D}]$
**for** $N_{iter}$ *iterations* **do**
    **for** $i = 1, \cdots, T_{tr}$ **do**
        $g_{\text{GMSGD}}(\boldsymbol{\theta}_0, \mathbf{W}, \mathcal{X}_i, \mathcal{Y}_i) = \mathbf{W}^\top \mathbf{z}_i(\boldsymbol{\theta}_0)$   [Inner update]
    **end**
    $[\boldsymbol{\theta}_0, \mathbf{W}] = [\boldsymbol{\theta}_0, \mathbf{W}] - \beta \nabla_{[\boldsymbol{\theta}_0, \mathbf{W}]} \left( \sum_{i=1}^{T_{tr}} \mathcal{L}(\bar{\mathcal{X}}_i, \bar{\mathcal{Y}}_i, g_{\text{GMSGD}}(\boldsymbol{\theta}_0, \mathbf{W}, \mathcal{X}_i, \mathcal{Y}_i)) + \mu \Omega(\mathbf{W}) \right)$
**end**

---

## 3    TASK-SIMILARITY AWARE META-LEARNING

It is well known that the expressive power of linear regression is limited due to both its linear nature and the finite dimension of the input. Further, since the dimension of linear regression matrix $\mathbf{W}$ grows quadratically with the dimension of $\boldsymbol{\theta}$, a large amount of training data would be necessary to estimate it. A transformation of linear regression in the form of *'kernel substitution'* or *'kernel trick'* results in the more general nonparametric or kernel regression (Bishop, 2006; Schölkopf & Smola, 2002). Kernel regression essentially performs linear regression in an infinite dimensional space making it a simple yet powerful and effective nonlinear approach. This motivates us to use kernel regression model as the natural next step from the Generalized Meta-SGD developed in the earlier section. By generalizing the linear regression model in $g_{\text{GMSGD}}$, we propose an adaptation function $\mathbb{R}^D \times \mathbb{R}^{T_{tr} \times D} \times \mathbb{R}^{K n_x} \times \mathbb{R}^{K n_y} \ni (\boldsymbol{\theta}_0, \boldsymbol{\Psi}, \mathcal{X}, \mathcal{Y}) \mapsto g_{\text{TANML}}(\boldsymbol{\theta}_0, \boldsymbol{\Psi}, \mathcal{X}, \mathcal{Y}) \in \mathbb{R}^D$ in the form of nonparametric or kernel regression model given by

$$g_{\text{TANML}}(\boldsymbol{\theta}_0, \boldsymbol{\Psi}, \mathcal{X}_i, \mathcal{Y}_i) = \sum_{j=1}^{T_{tr}} \boldsymbol{\psi}_j k(\mathbf{z}_i(\boldsymbol{\theta}_0), \mathbf{z}_j(\boldsymbol{\theta}_0)) = \boldsymbol{\Psi}^\top \mathbf{k}(\boldsymbol{\theta}_0, i), \tag{2}$$

where $k : \mathbb{R}^{2D} \times \mathbb{R}^{2D} \mapsto \mathbb{R}$ denotes a valid kernel function[1], $\mathbf{k}(\boldsymbol{\theta}_0, i) \triangleq \left[ k\left(\mathbf{z}_i(\boldsymbol{\theta}_0), \mathbf{z}_1(\boldsymbol{\theta}_0)\right), \cdots, k\left(\mathbf{z}_i(\boldsymbol{\theta}_0), \mathbf{z}_{T_{tr}}(\boldsymbol{\theta}_0)\right) \right]^\top$ is the vector with kernel values between the $i$th training task and all the $T_{tr}$ training tasks, and $\boldsymbol{\Psi} = [\boldsymbol{\psi}_1, \cdots, \boldsymbol{\psi}_{T_{tr}}]$ is the matrix of kernel regression coefficients stacked along the columns. The kernel coefficient matrix $\boldsymbol{\Psi}$ and the parameter $\boldsymbol{\theta}_0$ are learnt in the meta-training phase by iteratively performing an outer update as in the case of the Generalized Meta-SGD. The computed $\boldsymbol{\Psi}$ and $\boldsymbol{\theta}_0$ are then used to train the predictor for a new test task by applying the inner update on its training data. We call our approach *Task-similarity Aware Nonparametric Meta-Learning (TANML)* since the kernel measures the similarity between tasks through the task-descriptors. As we show in the Appendix, TANML reduces to Meta-SGD and MAML for specific choices of the kernel $k$ and the regression coefficient matrix $\boldsymbol{\psi}$.

The kernel regression in equation 2 models $\boldsymbol{\theta}_i$ as belonging to the space of functions defined as $\mathcal{H}$:

$$\mathcal{H} = \left\{ \tilde{\boldsymbol{\theta}} : \tilde{\boldsymbol{\theta}} = \sum_{i'=1}^{T_{tr}} \tilde{\boldsymbol{\psi}}_i k(\mathbf{z}_i(\boldsymbol{\theta}_0), \mathbf{z}_{i'}(\boldsymbol{\theta}_0)), \; \tilde{\boldsymbol{\psi}}_{i'} \in \mathbb{R}^D, \; i' = 1, \cdots, T_{tr} \right\} \tag{3}$$

The space $\mathcal{H}$ is referred to as the *reproducing kernel Hilbert space* (RKHS) associated with the kernel $k(\cdot, \cdot)$. We refer the reader to (Hofmann et al., 2008) and (Schölkopf & Smola, 2002) for further reading on RKHS. The space $\mathcal{H}$ has an important structure: each function in $\mathcal{H}$ uses the

---

[1]A valid kernel function is one that results in the kernel matrix evaluated for any number of datapoints to be symmetric and positive-semidefinite cf. (Bishop, 2006)

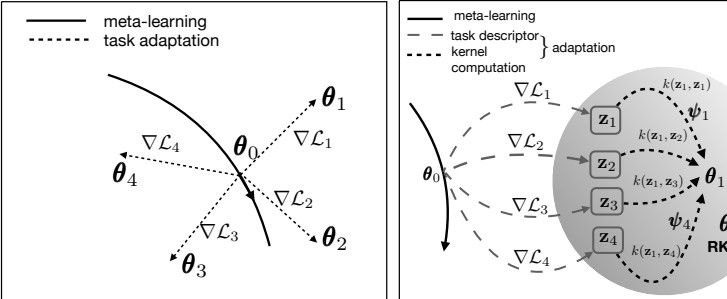

Figure 1: *Left:* Schematic of MAML. *Right:* Schematic of the TANML. Only the computation of $\boldsymbol{\theta}_1$ is shown to keep the diagram uncluttered.

information (the coefficients $\bar{\boldsymbol{\psi}}$) from all the $T_{tr}$ training tasks weighted by the kernel that essentially quantifies a similarity or correlation between the tasks through the task-descriptors defined earlier. While all meta-learning approaches use similarity implicitly, the number of works actively using similarity with training data at test time are limited (Fakoor et al., 2020). Computing the optimal values of the kernel coefficients $\boldsymbol{\Psi}$ and $\boldsymbol{\theta}_0$, which forms the meta-training phase, is then equivalent to solving the functional minimization problem:

$$\arg \min_{\boldsymbol{\theta}_0, \tilde{\boldsymbol{\theta}} \in \mathcal{H}} \left( \sum_{i=1}^{T_{tr}} \mathcal{L}(\bar{\mathcal{X}}_i, \bar{\mathcal{Y}}_i, \tilde{\boldsymbol{\theta}}) + \mu \|\tilde{\boldsymbol{\theta}}\|_{\mathcal{H}}^2 \right),$$

where the regularization term is the squared-norm in the RKHS which promotes smoothness and controls overfitting, $\mu$ being the regularization constant. The squared-norm in an RKHS is defined as $\|\tilde{\boldsymbol{\theta}}\|_{\mathcal{H}}^2 \triangleq \sum_{i=1}^{T_{tr}} \sum_{i'=1}^{T_{tr}} \boldsymbol{\psi}_i \boldsymbol{\psi}_{i'} k(\mathbf{z}_i(\boldsymbol{\theta}_0), \mathbf{z}_{i'}(\boldsymbol{\theta}_0)) = \boldsymbol{\Psi}^\top \mathbf{K}(\boldsymbol{\theta}_0) \boldsymbol{\Psi}$; and $\mathbf{K}(\boldsymbol{\theta}_0) \in \mathbb{R}^{T_{tr} \times T_{tr}}$ is the matrix of kernels evaluated across all the training tasks. This novel connection between meta-learning and RKHS obtained from TANML could potentially help in the mathematical understanding of existing algorithms, and help develop new meta-learning algorithms in the light of the RKHS theory (Hofmann et al., 2008; Schölkopf & Smola, 2002).

The meta-training phase for TANML is described in Algorithm 4, where we use $\Omega(\boldsymbol{\Psi}) = \boldsymbol{\Psi}^\top \mathbf{K}(\boldsymbol{\theta}_0) \boldsymbol{\Psi}$. In general, other regularizations such as $\ell_1$ or $\ell_2$ norms could also be used. We also note from equation 2 that the *TANML approach is a general framework*: any kernel and any task-descriptor which meaningfully captures the information in the task could be employed. What constitutes a meaningful descriptor for a task is an open question; while there have been studies on deriving features and metrics for understanding the notion of similarity between data sources or datasets, they have mostly been domain-specific and often require separate 'probe' neural networks for the extraction of features (Kim et al., 2019). The particular form of the task-descriptors used in our derivation is the result of taking MAML/Meta-SGD as a starting point, and follows naturally from analyzing them through the lens of linear and kernel regression. A schematic describing the task-descriptor based TANML and the intuition behind its working is shown in Figure 1.

---

**Algorithm 4:** Task-similarity Aware Meta Learning

---

Initialize $[\boldsymbol{\theta}_0, \boldsymbol{\Psi} \in \mathbb{R}^{T_{tr} \times D}]$
**for** $N_{iter}$ *iterations* **do**
    **for** $i = 1, \cdots, T_{tr}$ **do**
        $g_{\text{TANML}}(\boldsymbol{\theta}_0, \boldsymbol{\Psi}, \mathcal{X}_i, \mathcal{Y}_i) = \boldsymbol{\Psi}^\top \mathbf{k}(\boldsymbol{\theta}_0, i)$   [Inner update]
    **end**
    $[\boldsymbol{\theta}_0, \boldsymbol{\Psi}] = [\boldsymbol{\theta}_0, \boldsymbol{\Psi}] - \beta \nabla_{[\boldsymbol{\theta}_0, \boldsymbol{\Psi}]} \sum_{i=1}^{T_{tr}} \mathcal{L}(\bar{\mathcal{X}}_i, \bar{\mathcal{Y}}_i, g_{\text{TANML}}(\boldsymbol{\theta}_0, \boldsymbol{\Psi}, \mathcal{X}_i, \mathcal{Y}_i)) + \mu \Omega(\boldsymbol{\Psi})$   [Outer update]
**end**

---

**On the choice of kernels and sequential training** While the expressive power of kernels is immense, it is also known that the performance could vary depending on the choice of the kernel function(Schölkopf & Smola, 2002). The kernel function that works best for a dataset is usually found by trial and error. A possible approach is to use multi-kernel regression where one lets the data decide which of the pre-specified set of kernels are relevant (Sonnenburg & Schäfer, 2005;

Gönen & Alpaydin, 2011). Domain-specific knowledge may also be incorporated in the choice of kernels. In our analysis, we use two of the popular kernel functions: the Gaussian or the radial basis function (RBF) kernel, and the cosine kernel.

We note that since MAML and Meta-SGD and similar approaches perform the inner update independently for every task, they naturally admit a sequential or batch based training. Since TANML uses an inner update in the form of a nonparametric kernel regression, it inherits one of the limitations of kernel-based approaches − that all training data is used simultaneously for every task. As a result, the task losses and the associated gradients for all the training tasks are used at every inner update of TANML. One way to overcome this limitation would be the use of online or sequential kernel regression techniques (Lu et al., 2016; Sahoo et al., 2019; Vermaak et al., 2003). We will pursue this in our future work.

## 4 NUMERICAL EXPERIMENTS

We evaluate the performance of TANML and compare it with MAML and Meta-SGD on two synthesized regression datasets, five real-world time-series prediction problems from the Physionet 2012 Challenge dataset (Silva et al., 2012; Rothfuss et al., 2020), and on few-shot classification dataset Omniglot (Lake et al., 2015). The synthesized regression tasks have been used previously by previous works (Denevi et al., 2018; Finn et al., 2017; 2018) in meta-learning as a baseline for evaluating the performance on regression tasks. In every experiment, the predictor $f(x, \boldsymbol{\theta})$ is the output of a fully-connected four-layer feed-forward neural network, with Rectified linear unit (ReLU) as the activation function; $\boldsymbol{\theta}$ is the vector of all the weights and biases in the neural network - the predicted output is a scalar $\hat{\mathsf{f}}$ or vector $\hat{\mathbf{y}}$, for the regression and classification tasks, respectively. We consider two kernel functions for TANML: the Gaussian kernel $k(\mathbf{z}_i(\boldsymbol{\theta}_0), \mathbf{z}_{i'}(\boldsymbol{\theta}_0)) = \exp\left(-\|\mathbf{z}_i(\boldsymbol{\theta}_0) - \mathbf{z}_{i'}(\boldsymbol{\theta}_0)\|_2^2/\sigma^2\right)$, and the cosine kernel $k(\mathbf{z}_i(\boldsymbol{\theta}_0), \mathbf{z}_{i'}(\boldsymbol{\theta}_0)) = \frac{\mathbf{z}_i(\boldsymbol{\theta}_0)^\top \mathbf{z}_{i'}(\boldsymbol{\theta}_0)}{\|\mathbf{z}_i(\boldsymbol{\theta}_0)\|\|\mathbf{z}_{i'}(\boldsymbol{\theta}_0)\|}$. In order that the structural similarities are better expressed, we update kernel regression for the parameters of the different layers separately. This is because using a single adaptation function all components of $\boldsymbol{\theta}_i$ might result in certain parameters dominating the kernel regression, specially when the dimension of the parameters becomes large. Hence, we perform the adaptation separately for components of $\boldsymbol{\theta}_i$ corresponding to the different layers $l = 1, \cdots, L$ using the adaptation functions $\mathbb{R}^{D_l} \times \mathbb{R}^{D_l \times T_{tr}} \mathbb{R}^{K n_x} \times \mathbb{R}^{K n_y} \ni (\boldsymbol{\theta}_{0,l}, \boldsymbol{\Psi}_l, \mathcal{X}, \mathcal{Y}) \mapsto g_{\text{TANML},l}(\boldsymbol{\theta}_{0,l}, \boldsymbol{\Psi}_l, \mathcal{X}, \mathcal{Y}) \in \mathbb{R}^{D_l}$ for the parameter $\boldsymbol{\theta}_{i,l}$ belonging to the $l$th network layer: $\boldsymbol{\theta}_{i,l} = g_{\text{TANML},l}(\boldsymbol{\theta}_{0,l}, \boldsymbol{\Psi}_l, \mathcal{X}_i, \mathcal{Y}_i) = \sum_{i'=1}^{T_{tr}} \boldsymbol{\psi}_{i',l} k(\mathbf{z}_i(\boldsymbol{\theta}_{0,B}), \mathbf{z}_{i'}(\boldsymbol{\theta}_{0,l}))$, $l = 1, \cdots, L$. The NMSE performance on the meta-test set is obtained by averaging over 30 Monte Carlo realizations of tasks. For the regression tasks, the performance of the various meta-learning approaches are compared using the normalized mean-squared error (NMSE) on the test tasks: $\text{NMSE} \triangleq \frac{\sum_{i=1}^{T_v} \sum_{k=1}^{K} (y_k - \hat{y}_k)^2}{\sum_{i=1}^{T_v} \sum_{k=1}^{K} y_k^2}$. The numerical details of the experiments not mentioned in the manuscript, such as the learning rate and other hyper-parameters, are given in the appendix for space constraints.

**Experiment 1** We consider the task of training linear predictors of the form $f(x, \boldsymbol{\theta}) = \boldsymbol{\theta}^\top x$. The task data pairs $(x, y) \in \mathbb{R}^{16} \times \mathbb{R}$ are generated by a linear model $y = \beta^\top x + e$. The regression coefficient vector $\beta$ for different tasks is randomly sampled with equal probability from two isotropic Gaussian distributions on $\beta$: with means $\beta_0 = -\mathbf{4}$ and $\beta_0 = \mathbf{4}$, where $\mathbf{4}$ denotes the vector of all fours. The additive noise $e$ is assumed to be white and drawn from the standard normal distribution and uncorrelated with $x$, which is distributed as the multivariate normal distribution of size 16. We consider two cases of $T_{tr} = 32$ and $T_{tr} = 64$ training tasks, Each task with $K = 4$ samples in training and test sets. We evaluate the performance of the MAML, Meta-SGD, and TANML on a test set of $T_v = 64$ tasks. The NMSE test performance is reported in Table 1. We observe that both MAML and Meta-SGD perform very poorly in comparison to TANML; Meta-SGD performs slightly better than MAML. Further, we observe that TANML with the Cosine kernel performs the best among the four algorithms. The superior performance of TANML could be ascribed its the nonlinear nature with the gradients enter the estimation through the kernels. The adaptation function involves terms with products of different gradients acting in the spirit of a higher order method unlike MAML/Meta-SGD that use a first order adaptation. This also corroborates with the findings of the recent theoretical work by Saunshi et al. (2020), where they show that MAML-type approaches can fail under convex settings (as is the case in this experiment). It is also interesting to

| Algorithm | MAML | Meta-SGD | TANML-Gaussian | TANML-Cosine |
|---|---|---|---|---|
| $T_{tr} = 32$ | 0.95 | 0.91 | **0.185** | **0.079** |
| $T_{tr} = 64$ | 0.91 | 0.86 | **0.15** | **0.070** |

Table 1: NMSE on test tasks for the regression experiment 1.

| Algorithm | Experiment 2a 10% outlier | | Experiment 2a 20% outlier | | Experiment 2b 10% outlier | | Experiment 2b 20% outlier | |
|---|---|---|---|---|---|---|---|---|
| $T_{tr}$ | 256 | 512 | 256 | 512 | 256 | 512 | 256 | 512 |
| MAML | 0.83 | 0.77 | 0.75 | 0.74 | 0.89 | 0.81 | 0.83 | 0.76 |
| Meta-SGD | 0.92 | 1.04 | 0.81 | 0.93 | 1.5 | 0.92 | 1.06 | 0.93 |
| **TANML-Gaussian** | **0.4** | **0.41** | **0.38** | **0.38** | **0.76** | **0.60** | **0.73** | **0.58** |
| **TANML-Cosine** | **0.37** | **0.35** | **0.30** | **0.26** | **0.44** | **0.38** | **0.47** | **0.33** |

Table 2: NMSE on test tasks for regression experiment 2.

note that the value of $\boldsymbol{\theta}_0$ we obtain for TANML almost coincides with $\mathbf{0}$, which is the value for $\boldsymbol{\theta}_0$ that theoretically minimizes the average error for this problem. We also find that both cosine and Gaussian kernels converge typically in about 5000 iterations, whereas MAML and Meta-SGD do not show improvement in NMSE even after 30000 iterations.

**Experiment 2** In this experiment, we consider the task of training of non-linear predictors which correspond to a fully connected ANN. We consider data pairs $(x, y) \in \mathbb{R} \times \mathbb{R}$ generated from the sinusoidal data source $y = A\sin(\omega x)$, where $x$ is drawn randomly from the interval $[-1, 1]$, and $A$ and $\omega$ differ across tasks. We do not use the knowledge that the data comes from a sinusoidal source while training the predictors. We are given $K = 4$ shots or data-pairs in each task. In order to illustrate the potential of TANML in using the similarity/ dissimilarity among tasks, we consider a fixed fraction of the tasks to be outliers, that is, generated from a non-sinusoidal data source in both meta-training and meta-test data, as described next. We consider two different regression experiments:

(1) *Experiment 2a − Fixed frequency varying amplitude:* The data for the different tasks generated from sinusoids with $A$ drawn randomly from $(0, 1]$, setting $\omega = 1$. The outlier task data generated as $y(x) = Ax$.
(2) *Experiment 2b − Fixed amplitude varying frequency:* The data for the different tasks generated from sinusoids with $\omega$ randomly drawn from $[1, 1.5]$, setting $A = 1$. The outlier task data is generated as $y(x) = \omega x$.

We perform the experiments with the number of meta-training tasks equal to $T_{tr} = 256$ and $T_{tr} = 512$. The NMSE performance on test tasks obtained by averaging over 100 Monte Carlo realizations of tasks is reported Table . We observe that TANML outperforms both MAML and Meta-SGD in test prediction by a significant margin even when the fraction of the outlier tasks is 10% and 20%. This clearly supports our intuition that an explicit awareness or notion of similarity aids in the learning, specially when the number of training tasks is limited. We also observe that on an average TANML with the cosine kernel performs better than the Gaussian kernel. We note that the performance of the approaches in Experiment 2a is better than that in Experiment 2b. This is because there is higher variation among the tasks (changing frequency) than in Experiment 2a (changing amplitudes). We also observe that the performance improves slightly as $T_{tr}$ increases.

**Experiment 3** In this experiment, we consider the the regression task of time-series prediction on the ICU patient dataset from the Physionet 2012 challenge. The dataset consists of measurements of various vital characteristics of different patients monitored over a period of 48 hours, logged in at various non-uniform time instants. For a given vital characteristic, the time-series of every individual patient forms a task: the time of measurement and the measured value are the input and output, respectively. The goal of the experiment is to predict the time series for new unseen patients given the measurements taken during the first 24 hours. The experiment is done in five different experiment settings corresponding to five different vital characteristics($V_1$ to $V_5$ ). Since our interest is in evaluating the performance in the limited task setting, we consider 100 tasks each for the meta-training, meta-validation, and meta-testing. Further details of the implementation are given in the Appendix. The NMSE values obtained for the test tasks are shown in Table 3. We observe that in all the experiments TANML with the Gaussian kernel results in the least NMSE, though we also note

| Algorithm | $V_1$ | $V_2$ | $V_3$ | $V_4$ | $V_5$ |
|---|---|---|---|---|---|
| MAML | 0.56 | 0.50 | 0.12 | 0.13 | 1.2 |
| Meta-SGD | 0.03 | 0.18 | 0.04 | 0.03 | 0.63 |
| TANML-Cosine | 0.13 | 0.25 | 0.05 | 0.06 | 0.56 |
| **TANML-Gaussian** | **0.03** | **0.12** | **0.04** | **0.02** | **0.42** |

Table 3: NMSE on test tasks for regression experiment on Physionet 2012 dataset.

| Algorithm | Experiment 4a: $T_{tr} = 50$ | Experiment 4b: $T_{tr} = 150$ |
|---|---|---|
| MAML | 16 | 18.2 |
| Meta-SGD | 18.4 | 19.5 |
| **TANML-Gaussian** | **19.6** | **22** |
| **TANML-Cosine** | **23.2** | **24.1** |

Table 4: Test accuracy % on Omniglot dataset.

that the performance of Meta-SGD also coincides with that of TANML in some of the experiments. On comparison with results from Experiment 2, we observe that while Meta-SGD shows significant improvement over MAML, it exhibits severe instability in the presence of outliers. On the other hand, we observe that TANML performs good prediction in both the experiments.

**Ablation Study**  An ablation study of some aspects of TANML is given in Section B of the Appendix. The study is made on the data from Experiment 3.

**Few-shot learning**  We now consider the application of our approach to few-shot learning on the character classification data from the Omniglot dataset. We consider the case of 5-way 1-shot learning: that is each task consists of a 5-class problem where each class is given two samples: one for training and the other for validation. The classes correspond to different written characters /symbols of a language, the input is the image in a gray-scale value. Given one training image per class, the goal of each task is to build a 5-way classifier and evaluate its performance on another set of five images. Unlike the state-of-the-art approaches where the goal is to perform few-shot learning over a large number of training tasks (Li et al., 2017), we restrict our experiments to the limited task setting. We consider two settings, where we use $T_{tr} = 50$ and $T_{tr} = 100$ meta-training tasks, and evaluate the performance on a test set of 100 tasks. The cross-entropy loss is used as the objective function for training. The test performance is measured in terms of the accuracy defined as the ratio of correctly classified samples to the total number of samples in each task and is reported in Table 4. We observe that TANML with both the cosine and the Gaussian kernel outperform MAML and Meta-SGD. We note that the accuracy values are low due to the limited task setting that we have considered. Nevertheless, the experiments demonstrate the potential of TANML to improve few-shot learning in extremely data-limited scenarios. We predict a similar trend to be exhibited even when the number of training tasks is large.

## 5  CONCLUSION

We proposed a task-similarity aware meta-learning algorithm that explicitly quantifies and employs a similarity between tasks through nonparametric kernel regression. We showed how our approach brings a novel connection between meta-learning and reproducing kernel Hilbert spaces. Our hypothesis was that an explicit incorporation of task-similarity helps improve the meta-learning performance in the task-limited setting with possible outlier tasks. Experiments with regression and classification tasks support our hypothesis, and our algorithm was shown to outperform the popular meta-learning algorithms by a significant margin. The aim of the current contribution was to investigate how task-similarity could be meaningfully employed and used to advantage in meta-learning. To that end, we wish to reiterate that the study is an ongoing one and the experiments considered in this paper are in no way exhaustive. An important next step for our approach is also the use of online/sequential kernel regression techniques to run our algorithm in a sequential or batch-based manner and scale to scenarios with large number of tasks. The nonparametric kernel regression framework also opens doors to a probablistic or Bayesian treatment of meta-learning that we plan to pursue in the recent future.

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

## A  CONNECTION OF TANML TO META-SGD AND MAML

We note that in the case when the kernel is the linear kernel $k(\mathbf{z}_i(\boldsymbol{\theta}_0), \mathbf{z}_{i'}(\boldsymbol{\theta}_0)) = \mathbf{z}_i(\boldsymbol{\theta}_0)^\top \mathbf{z}_{i'}(\boldsymbol{\theta}_0)$, TANML becomes the special case of Generalized Meta-SGD proposed in Section 2.2. This is because kernel regression and GMSGD further reduces to the Meta-SGD when the linear regression matrix $\mathbf{W}$ is constrained to be of the form as in equation 1 and the regularization $\mu$ in the outer loop is set to zero. When regression coefficients of the GMSGD are fixed to $\mathbf{W} = [\mathbf{I}, -\alpha\mathbf{I}]$, we obtain the MAML, which is a fixed linear transform acting on the task-descriptor. Thus, TANML reduces to the specific cases of Meta-SGD and MAML, when the kernel is the linear kernel and the regularization parameter $\mu$ in the outer-loop is set to zero.

Let us again consider the adaptation rule of the TANML:

$$g_{\text{TANML}}(\boldsymbol{\theta}_0, \boldsymbol{\Psi}, \mathcal{X}, \mathcal{Y}) = \sum_{j=1}^{T_{tr}} \boldsymbol{\psi}_j k(\mathbf{z}_i(\boldsymbol{\theta}_0), \mathbf{z}_j(\boldsymbol{\theta}_0)) = \boldsymbol{\Psi}^\top \mathbf{k}(\boldsymbol{\theta}_0, i)$$

In the case of linear kernel, we have that $k(\mathbf{z}_i(\boldsymbol{\theta}_0), \mathbf{z}_j(\boldsymbol{\theta}_0)) = \mathbf{z}_i(\boldsymbol{\theta}_0)^\top \mathbf{z}_j(\boldsymbol{\theta}_0)$. Then, we have that

$$\mathbf{k}(\boldsymbol{\theta}_0, i) = \mathbf{Z}^\top(\boldsymbol{\theta}_0) z(\boldsymbol{\theta}_0, i)$$

where $\mathbf{Z}(\boldsymbol{\theta}_0)$ is the matrix of the task descriptors of all the training tasks arranged column-wise. This gives that

$$g_{\text{TANML}}(\boldsymbol{\theta}_0, \boldsymbol{\Psi}, \mathcal{X}, \mathcal{Y}) \quad = \quad \boldsymbol{\Psi}^\top \mathbf{k}(\boldsymbol{\theta}_0, i) = \boldsymbol{\Psi}^\top \mathbf{Z}^\top(\boldsymbol{\theta}_0) z(\boldsymbol{\theta}_0, i)$$

In the case when $\boldsymbol{\Psi}^\top \mathbf{Z}(\boldsymbol{\theta}_0)^\top = [\mathbf{I} - \text{diag}(\boldsymbol{\alpha})]$, we get that

$$g_{\text{TANML}}(\boldsymbol{\theta}_0, \boldsymbol{\Psi}, \mathcal{X}, \mathcal{Y}) = \boldsymbol{\Psi}^\top \mathbf{Z}^\top(\boldsymbol{\theta}_0) z(\boldsymbol{\theta}_0, i) = [\mathbf{I} - \text{diag}(\boldsymbol{\alpha})] z(\boldsymbol{\theta}_0, i) = g_{\text{MSGD}}$$

That is, when $\boldsymbol{\Psi} = (\mathbf{Z}(\boldsymbol{\theta}_0))^\dagger [\mathbf{I} - \text{diag}(\alpha)]^\top$, $g_{\text{TANML}} = g_{\text{MSGD}}$, where $\dagger$ denotes the pseudo-inverse operation. In other words, TANML with the kernel regression matrix constrained to be of the form $\boldsymbol{\Psi} = (\mathbf{Z}(\boldsymbol{\theta}_0))^\dagger [\mathbf{I} - \text{diag}(\alpha)]^\top$ and with no regularization or $\mu$ reduces to the Meta-SGD.

Further, by the same argument, we have that in the case when TANML with $\boldsymbol{\Psi}$ given by the fixed matrix $\boldsymbol{\Psi} = (\mathbf{Z}(\boldsymbol{\theta}_0))^\dagger [\mathbf{I} - \alpha\mathbf{I}]^\top$, gives

$$g_{\text{TANML}} = g_{\text{MSGD}}$$

This also helps gives an intuition on why behaviour of the TANML might perform better − MAML and Meta-SGD are obtained by restricting the set of possible $\boldsymbol{\Psi}$ to a constrained set. In contrast, TANML obtains $\boldsymbol{\Psi}$ through unconstrained search over $\mathbb{R}^{T_{tr} \times D}$.

Thus, MAML and Meta-SGD are obtained as special cases of TANML, when the kernel is the linear or the simple inner-product kernel, and the regression matrix $\boldsymbol{\Psi}$ takes special forms.

## B  ABLATION STUDY

We consider the study of the influence of two aspects on TANML:

### B.1  REGULARIZATION PARAMETER $\mu$

As in the case of any regularized estimation or modelling approach, the motivation of including the regularization parameter $\mu$ is to avoid the overfitting for $\boldsymbol{\psi}$, specially in the task-limited setting. Keeping all other parameters unchanged, we vary $\mu$ and the resulting test NMSE is shown in Table 5.

We observe that both a very large $\mu$ or a very small $\mu$ degrades the test performance. This is because unlike MAML and Meta-SGD, the since TANML estimates more number of parameters from the same training data, it tends to quickly overfit the training data to almost zero error. Hence, a nonzero value of $\mu$ helps curtail the overfit. However, setting $\mu$ to a large value biases the parameters such that very little learning takes place beyond the initial few meta-iterations resulting also in a poor test performance. Thus, we see the best test prediction being acheived with $\mu$ set to $0.01$.

| $\mu$ | $V_1$ Cosine | $V_1$ Gaussian | $V_3$ Cosine | $V_3$ Gaussian | $V_4$ Cosine | $V_4$ Gaussian |
|---|---|---|---|---|---|---|
| 5 | 0.63 | 0.56 | 0.39 | 0.31 | 0.52 | 0.63 |
| 0.1 | 0.34 | 0.08 | 0.06 | 0.04 | 0.05 | 0.05 |
| **0.01** | **0.13** | **0.03** | **0.05** | **0.04** | **0.06** | **0.02** |
| 0.001 | 0.32 | 0.08 | 0.1 | 0.07 | 0.08 | 0.06 |

Table 5: Effect of $\mu$ for regression experiment on Physionet 2012 dataset for Ex1, Ex2, Ex3.

This in turn indicates that the regularization helps TANML perform better than MAML and Meta-SGD by avoiding overfitting to the training data. This is perhaps why TANML exhibits a superior performance in comparison with the other two methods − as we have seen in Section A of the Appendix, MAML and Meta-SGD are obtained when TANML uses a linear kernel and zero regularization.

## C    NUMERICAL EXPERIMENTS

We compare four different approaches: MAML, Meta-SGD, TANML-Cosine, TANML-Gaussian. All the algorithms were trained for 60000 meta-iterations, where each meta-iteration outer update uses the entire set of training tasks, and not as a stochastic gradient descent. All the experiments were performed on either NVIDIA Tesla K80 GPU on Microsoft Azure Platform.

## D    EXPERIMENT 1 HYPERPARAMETERS

In this experiment, we use a linear predictor of dimension 16, same as the input dimension. Each task has four training input-output pairs and 4 test input-output pairs, that is, $K = 4$. Input $x \in \mathbb{R}^{16}$ is drawn from $\mathcal{N}(0, \mathbf{I})$, the additive noise $e$ is white and drawn from $\mathcal{N}(0, 1)$.

### D.1    MAML

- Inner update learning rate: $\alpha$: 0.01
- Outer update learning rate: $5 \times 10^{-4}$
- Optimizer: Adam

### D.2    META-SGD

- Inner update learning rate $\boldsymbol{\alpha}$: learnt, initialized with values randomly drawn from $[0.001, 0.01]$
- Outer update learning rate for $\boldsymbol{\theta}_0$: $5 \times 10^{-4}$
- Outer update learning rate for $\boldsymbol{\alpha}$: $1 \times 10^{-6}$
- Optimizer: Adam

### D.3    TANML-GAUSSIAN

- Outer update learning rate for $\boldsymbol{\theta}_0$: $1 \times 10^{-3}$
- Outer update learning rate for $\boldsymbol{\Psi}$: $5 \times 10^{-5}$
- $\mu = 0.1$
- $\sigma^2 = 0.5$
- Optimizer: Adam

### D.4    TANML-COSINE

- Outer update learning rate for $\boldsymbol{\theta}_0 = \boldsymbol{\theta}_0$: $5 \times 10^{-4}$
- Outer update learning rate for $\boldsymbol{\Psi}$: $1 \times 10^{-5}$
- $\mu = 0.1$
- Optimizer: Adam

### D.5    EXPERIMENT 2 HYPER-PARAMETERS

The hyper-parameters for the four approaches are listed below. The learning-rate parameters were chosen such that the training error converged without instability.

(1) *Experiment 2a − Fixed frequency varying amplitude:* The data for the different tasks generated from sinusoids with $A$ drawn randomly from $(0, 1]$, setting $\omega = 1$. The outlier task data generated as $y(x) = Ax$.

(2) *Experiment 2b − Fixed amplitude varying frequency:* The data for the different tasks generated from sinusoids with $\omega$ randomly drawn from $[1, 1.5]$, setting $A = 1$. The outlier task data is generated as $y(x) = \omega x$.

## D.6 MAML

- Inner update learning rate: $\alpha$: 0.01
- Outer update learning rate: $5 \times 10^{-4}$
- Total ANN layers: 4 with, 2 hidden layers
- Non-linearity: ReLU
- Optimizer: Adam

## D.7 META-SGD

- Inner update learning rate $\boldsymbol{\alpha}$: learnt, initialized with values randomly drawn from $[0.001, 0.01]$
- Outer update learning rate for $\boldsymbol{\theta}_0$: $5 \times 10^{-4}$
- Outer update learning rate for $\boldsymbol{\alpha}$: $1 \times 10^{-6}$
- Total ANN layers: 4 with, 2 hidden layers
- Non-linearity: ReLU
- Optimizer: Adam

## D.8 TANML-GAUSSIAN

- Outer update learning rate for $\boldsymbol{\theta}_0$: $1 \times 10^{-3}$
- Outer update learning rate for $\boldsymbol{\Psi}$: $5 \times 10^{-5}$
- $\mu = 0.1$
- $\sigma^2 = 0.5$
- Total ANN layers: 4 with, 2 hidden layers
- Non-linearity: ReLU
- Optimizer: Adam

## D.9 TANML-COSINE

- Outer update learning rate for $\boldsymbol{\theta}_0 = \boldsymbol{\theta}_0$: $5 \times 10^{-4}$
- Outer update learning rate for $\boldsymbol{\Psi}$: $1 \times 10^{-5}$
- $\mu = 0.1$
- Total ANN layers: 4 with, 2 hidden layers
- Non-linearity: ReLU
- Optimizer: Adam

# E EXPERIMENT 3 HYPERPARAMETERS

For every task, we use half of the samples for training data, and the remaining half for the validation. In the case of test tasks, the training set consists of samples from the first half of the time series-the goal is to predict the rest of the time series. In the case of training tasks, the training and validation sets are drawn randomly from the entire time-series. We consider only those patients which have at least four measurements. In order to minimize the dynamic range of the measurements, we divided all the true measurements by 150. The time was measured in minutes. The five vital characteristics considered are(Silva et al., 2012):

- $V_1$: Blood urea nitrogen
- $V_2$: Creatinine
- $V_3$: Invasive diastolic arterial blood pressure
- $V_4$: Urine

- $V_5$: Heart-rate

The number of datapoints in each task varies from 4 to 48. The tasks with number of datapoints less than four are not considered.

The following hyper-parameters settings were using in all the five vital characteristics.

### E.1 MAML

- Inner update learning rate: $\alpha$: $10^{-3}$
- Outer update learning rate: $5 \times 10^{-3}$
- Total ANN layers: 4 with, 2 hidden layers, 8 neurons per layer
- Non-linearity: ReLU
- Optimizer: Adam

### E.2 META-SGD

- Inner update learning rate $\boldsymbol{\alpha}$: learnt, initialized with values randomly drawn from $[0.001, 0.01]$
- Outer update learning rate for $\boldsymbol{\theta}_0$: $5 \times 10^{-3}$
- Outer update learning rate for $\boldsymbol{\alpha}$: $1 \times 10^{-5}$
- Total ANN layers: 4 with, 2 hidden layers, 8 neurons per layer
- Non-linearity: ReLU
- Optimizer: Adam

### E.3 TANML-GAUSSIAN

- Outer update learning rate for $\boldsymbol{\theta}_0$: $1 \times 10^{-3}$
- Outer update learning rate for $\boldsymbol{\Psi}$: $5 \times 10^{-5}$
- $\mu = 0.01$
- $\sigma^2 = 1$
- Total ANN layers: 4 with, 2 hidden layers
- Non-linearity: ReLU
- Optimizer: Adam

### E.4 TANML-COSINE

- Outer update learning rate for $\boldsymbol{\theta}_0 = \boldsymbol{\theta}_0$: $5 \times 10^{-4}$
- Outer update learning rate for $\boldsymbol{\Psi}$: $1 \times 10^{-5}$
- $\mu = 0.01$
- Total ANN layers: 4 with, 2 hidden layers
- Non-linearity: ReLU
- Optimizer: Adam

## F  EXPERIMENT 4 HYPERPARAMETERS

We use two separate sets from training and test tasks as considered by Finn & Levine (2018). The images are resized to $64 \times 64$ dimensions.

### F.1    MAML

- Inner update learning rate: $\alpha$: $10^{-3}$
- Outer update learning rate: $5 \times 10^{-3}$
- Total ANN layers: 4 with, 2 hidden layers, 8 neurons per layer
- Non-linearity: ReLU
- Optimizer: Adam

### F.2    META-SGD

- Inner update learning rate $\boldsymbol{\alpha}$: learnt, initialized with values randomly drawn from $[0.001, 0.01]$
- Outer update learning rate for $\boldsymbol{\theta}_0$: $5 \times 10^{-4}$
- Outer update learning rate for $\boldsymbol{\alpha}$: $1 \times 10^{-5}$
- Total ANN layers: 4 with, 2 hidden layers, 8 neurons per layer
- Non-linearity: ReLU
- Optimizer: Adam

### F.3    TANML-GAUSSIAN

- Outer update learning rate for $\boldsymbol{\theta}_0$: $1 \times 10^{-3}$
- Outer update learning rate for $\boldsymbol{\Psi}$: $5 \times 10^{-5}$
- $\mu = 0.01$
- $\sigma^2 = 1$
- Total ANN layers: 4 with, 2 hidden layers
- Non-linearity: ReLU
- Optimizer: Adam

### F.4    TANML-COSINE

- Outer update learning rate for $\boldsymbol{\theta}_0 = \boldsymbol{\theta}_0$: $5 \times 10^{-4}$
- Outer update learning rate for $\boldsymbol{\Psi}$: $1 \times 10^{-5}$
- $\mu = 0.01$
- Total ANN layers: 4 with, 2 hidden layers
- Non-linearity: ReLU
- Optimizer: Adam

