# OpenReview forum: "Task-similarity Aware Meta-learning through Nonparametric Kernel Regression"
_ICLR.cc/2021/Conference — Reject_

### Official Review · AnonReviewer2 · 2020-10-21
**Good work in progress but further work is needed, both theoretical and experimental**

**Rating:** 3
**Confidence:** 5

**Review:**

---- Update ----

I thank the authors for clarifications. I trust that the suggestions of all reviewers, taken together, provide substantial avenues for improving the work. However, at this point I must keep my score and encourage the authors to continue the work with the valuable honest feedback provided here.

---- Original Review ----

Summary:

The paper aims to formalize “task similarity” in meta-learning settings by making use of nonparametric kernel regression techniques; such similarity information is then proposed as a means to alleviate some of the current issues with meta-learning algorithms such as MAML/Meta-SGD, namely reliance on large sets of similar meta-training tasks. Experiments focus on standard toy regression tasks with the added meta-training data scarcity.



Strong points:
- Principled approach to a challenging and current problem of interest for the sub-field of meta-learning and beyond. However, formalization of meta-learning approaches in terms of the NTK is recent but not novel [see reference Wang 2020 in paper].
- Good work in progress, but the attempt to publish is premature.



Weak points:
- Very poor representation of relevant and conceptually similar recent work, see [1] for a comprehensive review, and specifically [2, 3, 4, 5, 6] for similar approaches.
- Inaccurate claims of novelty are made; they must be made more specific and put into context. For example, the claim that task descriptors have not been used in the design of meta-learning algorithms is false, see [5, 6]. That said, the current approach could be used to analyze such SOTA approaches and perhaps explain their performance.
- Meta-training data reuse across tasks at test time has been proposed previously, e.g. [7], so it is also not novel to this paper.
- Very weak experimental evidence. Please use some of the few-shot image classification datasets, or standard RL tasks available since MAML was published.
- Proposed method needs extensive approximations to scale up to more interesting problems.




Recommendation and Rationale:

I believe the paper should be rejected in current form, but I strongly encourage the authors to add more experimental data and submit to a workshop.



References:
[1] Meta-Learning in Neural Networks: A Survey
Timothy Hospedales, Antreas Antoniou, Paul Micaelli, Amos Storkey. https://arxiv.org/pdf/2004.05439.pdf
[2] Recasting Gradient-Based Meta-Learning as Hierarchical Bayes
Erin Grant, Chelsea Finn, Sergey Levine, Trevor Darrell, Thomas Griffiths. https://arxiv.org/abs/1801.08930
[3] Bayesian Model-Agnostic Meta-Learning. Jaesik Yoon, Taesup Kim, Ousmane Dia, Sungwoong Kim, Yoshua Bengio, Sungjin Ahn. https://papers.nips.cc/paper/7963-bayesian-model-agnostic-meta-learning.pdf
[4] Probabilistic Model-Agnostic Meta-Learning. Chelsea Finn, Kelvin Xu, Sergey Levine. http://papers.nips.cc/paper/8161-probabilistic-model-agnostic-meta-learning
[5] Meta-Learning with Latent Embedding Optimization. Andrei A. Rusu, Dushyant Rao, Jakub Sygnowski, Oriol Vinyals, Razvan Pascanu, Simon Osindero, Raia Hadsell. https://arxiv.org/abs/1807.05960
[6] Few-Shot Image Recognition by Predicting Parameters from Activations. Siyuan Qiao, Chenxi Liu, Wei Shen, Alan Yuille. https://arxiv.org/abs/1706.03466
[7] Meta-Q-Learning. Rasool Fakoor, Pratik Chaudhari, Stefano Soatto, Alexander J. Smola. https://arxiv.org/abs/1910.00125

---

> ### Author Response · Authors · 2020-11-25
> **Authors' response to AnonReviewer2**
>
> **Authors**
>
> We would like to thank the reviewer for evaluating our work and sharing their feedback.
>
> **Reviewer**
>
> _Principled approach to a challenging and current problem of interest for the sub-field of meta-learning and beyond. However, formalization of meta-learning approaches in terms of the NTK is recent but not novel [see reference Wang 2020 in paper]._
>
> _Good work in progress, but the attempt to publish is premature
> Very poor representation of relevant and conceptually similar recent work, see [1] for a comprehensive review, and specifically [2, 3, 4, 5, 6] for similar approaches._
>
> **Authors**
>
> Thank you for drawing our attention to these valuable references. We have now included them in the manuscript in the proper context. PWe note that the NTK and meta-learning work by Wang et al considers a kernel which comes from an assymptotic analysis considering very wide neural networks (in many cases requiring the network size to tend to infinity). Our approach differs from these works since we does not require such an asymptotic analysis. Our kernel approach also is not restricted to the use of neural networks as predictors and can be applied to all types of the predictors. Further, the NTK uses a specific form of the kernel, whereas our formulation allows for any valid kernel function. We have now included a discussion on this in Section 1.2.
>
> **Reviewer**
>
> _Inaccurate claims of novelty are made; they must be made more specific and put into context. For example, the claim that task descriptors have not been used in the design of meta-learning algorithms is false, see [5, 6]._
>
> **Authors**
> Thank you for the comment. We have modified the appropriate portions of the manuscript. Please see second paragraph on page 6.
>
> **Reviewer**
>
> _That said, the current approach could be used to analyze such SOTA approaches and perhaps explain their performance._
>
> **Authors**
>
> Thank you for the encouraging comment. We indeed are considering to work along this direction in the future.
>
> **Reviewer**
>
> _Meta-training data reuse across tasks at test time has been proposed previously, e.g. [7], so it is also not novel to this paper._
>
> **Authors**
>
> We agree, and do not claim that data reuse at test time has not been proposed previously. We have also cited the reference [7] pointed out by you.
>
> **Reviewer**
>
> _Very weak experimental evidence. Please use some of the few-shot image classification datasets, or standard RL tasks available since MAML was published._
>
> **Authors**
>
> Thank you for the comment. After taking all the reviews carefully into consideration, we have now included multiple new experiments on real-world regression tasks and on few-shot learning for the Omniglot dataset. Please see the newly added paragraphs on Experiments 3 and 4 in Section 4 of the revised manuscript. We have also included new sections discussing some aspects of our approach, and its explicit relation to MAML and Meta-SGD in the Appendix
>
> **Reviewer**
>
> _Proposed method needs extensive approximations to scale up to more interesting problems._
>
> **Authors**
>
> We fully agree. As discussed in the manuscript earlier, the framework of kernel regression requires that all tasks are taken together and not sequentially, making it necessary to have approximations at higher dimensions. However, our focus was on investigating the merit of using task-similarity in the limited task setting. As a result, no approximations were found necessary or used in our experiments in this setting.

---

### Official Review · AnonReviewer1 · 2020-10-26
**Simple idea, results only in toy settings**

**Rating:** 4
**Confidence:** 4

**Review:**

The paper introduced a meta-learning framework in which a kernel describing similarity between the tasks is used to construct an RKHS which is used to perform kernel regression. The framework is instantiated in a form of an algorithm: TANML which can be viewed as an extension to a popular Meta-SGD algorithm. The experiments on two regression tasks are presented to analyse the efficacy of the proposed method.

1. I consider the method mathematically sound, ie. I don't see theoretical reasons which would make it obvious that it wouldn't work.
2. The combination of using task-similarity and kernels is not present in the literature known to me, so the work under review contains (some elements of) novelty.
3. However, both "explicitly employing task-similarity" (Achille et al. 2019) and "using kernel methods" (Vinyals et al. 2016) (separately) is well represented in past works.
4. In my opinion moving kernels from space of images/classes (like in Vinyals and other kernel methods cited in the paper) to the space of tasks doesn't, on its own, demonstrate the level of novelty that is required by accepted papers. Comparison to and commentary on the previous metric-based meta-learning papers present in the work under review is unsatisfactory.
5. The performed experiments are extremely toyish: the results are not sufficient to support the claims of the paper. This is further exacerbated by the fact that authors need to settle with optimization tricks to learn their TANML model.

I recommend against publication at ICLR. The novelty of the authors' work is limited and the experiments are not convincing. I appreciate that the goal of the work is not to beat SOTA, and rather to introduce and investigate a small change to MAML, but I believe that reducing the scope of the research in this way grants an expectation of extensive experimental evidence which this paper is lacking.

Suggestion:
To expand the research to higher-dimensional, few-shot classification tasks, one could hardcode the kernel based on the human understanding of the classes. This way, one could take a problem which Meta-SGD is known to be working with and see if an introduction of a kernel gives a measurable improvement.

Technical:
1. "Training for tasks individually will result in a predictor that overfits to $\mathcal{X}$, $\mathcal{Y}$, and generalizes poorly": It's unclear what "individually" means here (is MAML with batch size = 1 training for tasks individually?). It's also far from obvious, in particular in the context of Raghu et al. (Rapid Learning or Feature Reuse?). I encourage authors to avoid using such statements without referring to argumentation behind them.
2. I am also confused by the sentence "while there have been studies on deriving features and metrics for understanding the notion of similarity between data sources or datasets, they have not been used in the actual design of meta-algorithms": isn't the (cited by authors) Achille et al. (2019) one example of such work?
3. Along the whole paper, \citep is used, even when \citet is appropriate. See when to use each one [here](https://www.reddit.com/r/LaTeX/comments/5g9kn1/whats_the_difference_between_cite_citep_and_citep/).
4. TA*N*ML in Sec. 4.
5. .. realizations of tasks is reported **in** Table **2**. In Sec. 4.2.

---

> ### Author Response · Authors · 2020-11-25
> **Authors' response to AnonReviewer1**
>
> **Authors**
>
> We would like to thank the reviewer for evaluating our work and sharing their feedback.
>
> **Reviewer**
>
> _In my opinion... Comparison to and commentary on the previous metric-based meta-learning papers present in the work under review is unsatisfactory_
>
> **Authors**
>
> To the best of our knowledge, most of the existing kernel and metric-learning based methods deal exclusively with classification and image recognition/ few-shot learning settings, whereas our goal is to develop a general kernel-based formulation that is applicable to any meta-learning modality. In most cases, the use of kernel regression has been to describe similarity across the datapoints within a class or a task. In Achille et al. 2019, a separate and specific probe network is employed to extract features that are suited to visual classification tasks $-$ the task similarity is then used to select the best model from the existing set of models of the training tasks to describe the new task. This is different from our approach where parameters of a new task are obtained through similarity with training tasks, and not by setting the parameter value to that of one of the similar training tasks. Our task-descriptor comes directly from the task and model, without requiring an additional feature extraction network.
>
> We have now included additional relevant works in the related works, and also contrasted our approach with a more recent work on neural tangent kernels in meta-learning. Please see Section 1.2 of the revised manuscript. We agree that a more detailed comparison to existing metric learning approaches would throw further perspective and understanding of our approach and more generally on the use of metric-learning in meta-learning. We intend to pursue further research along these lines this in the future.
>
> **Reviewer**
>
> _The performed experiments are extremely toyish... authors need to settle with optimization tricks to learn their TANML model._
>
> **Authors**
>
> Thank you for the comment. We have now included additional experiments on real-world data for both regression and classification tasks. Please see the newly added paragraphs on Experiments 3 and 4 in Section 4 of the revised manuscript.
> Since we consider experiments in the regime of limited number of tasks in our experiments, no approximations or optimization tricks were involved or found necessary in learning our model.
>
> **Reviewer**
>
> _I recommend against publication at ICLR...an expectation of extensive experimental evidence which this paper is lacking._
>
> **Authors**
>
> We thank you for your valuable feedback. We have now included several new experiments on real-world regression tasks, and also on the Omniglot dataset. Please see the newly added paragraphs on Experiments 3 and 4 in Section 4 of the revised manuscript. We have also included new sections on ablation study of some aspects of our approach, and its explicit connections with Meta-SGD and MAML.  Please see Section A of the appendix.
>
> **Reviewer**
>
> _Suggestion: To expand the research to higher-dimensional,... and see if an introduction of a kernel gives a measurable improvement._
>
> **Authors**
>
> Thank you for the very insightful suggestion of incorporating of a hard-coded kernel with domain-specific intuition/understanding of tasks. Our work here presents a first step towards introduction of similarity kernels in meta-learning, our goal being to arrive at a consistent and general notion of kernel and task-descriptor. Hence, we opted the use of general kernels that appear from the formulation. However, we fully agree with the observation of the reviewer and in future work, we will pursue the use of alternative task-descriptors or kernels that aid in better scaling to higher dimensions through the incorporation of human understanding. This will include the use of domain-specific features/metrics as in the case of task2vec (Achille et al. 2019) and other recent works.
>
> **Reviewer**
>
> _"Training for tasks individually will result in a predictor that overfits to,... and generalizes poorly": It's unclear what "individually" means here...I encourage authors to avoid using such statements without referring to argumentation behind them._
>
> **Authors**
> Thank your for drawing our attention to this. The better word to describe is 'independently' and we have now changed it accordingly. Please see  revised Section 1.1.
>
> **Reviewer**
>
> _I am also confused by the sentence ...isn't the (cited by authors) Achille et al. (2019) one example of such work?_
>
> **Authors**
>
> Thank you for pointing this out. We have now modified the sentence. Please see the second paragraph on page 6.
>
> **Reviewer**
>
> _Along the whole paper, \citep is used, even when \citet is appropriate. See when to use each one here._
>
> **Authors**
>
> Thank you for pointing out the error. We have now modified the citation at the appropriate instances.
>
> **Reviewer**
>
> _TANML in Sec. 4._
>
> **Authors**
>
> Thank you, it is now rectified.

---

> > ### Comment · AnonReviewer1 · 2020-11-25
> > **Ok but not good enough**
> >
> > After reading the authors' response and skimming through the modified writeup, I keep my original score.
> >
> > Authors still don't compare to the previous, related work (merely list them).
> >
> > The addition of a simplified version of Omniglot (and another, "real-world" dataset) is a step in a good direction, yet still the experimental sophistication is behind MAML (which used full version of Omniglot and a harder dataset: Miniimagenet for few-shot classification), not to mention its contemporary, derived works.

---

### Official Review · AnonReviewer4 · 2020-10-27
**Overall a sound algorithm, but framed/motivated in a strange way and hardly supported by relevant/realistic experiments**

**Rating:** 4
**Confidence:** 4

**Review:**

=== Summary ===

The paper proposes a meta-learning method based on a notion of task similarity/dissimilarity. In particular, the paper motivates its proposed method TANML through a generalization of Meta-SGD wherein the learnable parameter-wise learning rate in the inner update of Meta-SGD is replaced by a quadratic pre-conditioner matrix.

The proposed method TANML closely resembles gradient-based meta-learners in the outer update but replaces the inner update by the matrix-vector product of kernel regression coefficient matrix and task similarity vector based on a kernel function. In that, the kernel function effectively quantifies the similarity of the loss gradients of the different tasks, evaluated at a learnable parameter initialization. Overall, the coefficient matrix can be understood as a look-up matrix in which each row holds the learned parameter vector for one meta-training task, the final adapted parameters are a linear combination of these parameter vectors in, weighted by the kernel between current task and the meta-train tasks.

In two simple simulated experiments, the paper demonstrates that TANML is able to outperform MAML and Meta-SGD when the meta-train tasks are set up in a pathological way (e.g. by combining two dissimilar clusters of tasks or by adding outlier tasks).

=== Reviewer’s main argument ===

Overall, the idea of incorporating a notion of task-similarity into the meta-learner and the particular proposal to use the kernel between the task loss gradients to quantify such similarity is sound and is a valuable contribution in itself.

Unfortunately, the relationship between Generalized Meta-SGD to TANML is unclear. Usually the connection between linear regression (c.f. Eq. 1 in the paper) and kernel regression (Eq. 2) is established through the particular form of the kernel regression coefficients. However, since the coefficient matrix is (meta-)learned in the paper, it is unclear how TANML relates to Meta-SGD. In fact, TANML seems more like a learned linear combination of task parameters which does not resemble much commonalities with MAML. Overall, the connection to MAML seems a bit set-up/artificial. Discussing the particular relationship between MAML/Meta-SGD and TANML would improve the storyline of the paper. For instance, if TANML is a generalization of MAML, it would be good to state with which particular choices of $\theta_0$ and $\Psi$, we can recover MAML.

The related work section is quite minimalistic. For instance, discussing how TANML is different from e.g. multi-task nonparametric methods (e.g. [1-2]) that also use a kernel between tasks, would better clarify how TANML relates to previous work.

The numerical experiments are very simple / limited and designed in a pathological way. Thus, it is not surprising that MAML/Meta-SGD perform worse than TANML. How applicable the experimental results are in more realistic meta-learning setups is unclear. Despite the simplicity of the experiments, there is not enough information to properly reproduce the experiment. For instance, how are A and $\omega$ in experiment 2 sampled, how are the x in experiment 1 sampled and how many data points per task are used in experiment 1? The following would strengthen the experiment section:
- A real-world use case in which we expect to see a meta-training set with e.g. outliers similar to experiment 2
- Experiments with real world meta-learning datasets. For real-world & small-scale meta-learning environments for regression, see e.g. [3].
- An additional meta-learning setup without outliers / clusters of meta-learning tasks. This way one can assess how the proposed method compares to MAML/Meta-SGD in standard setting
- Adding missing details, e.g. to the appendix, which are necessary for reproducing the experiment.

=== Overall assessment ===

I vote for rejecting the paper. In the current state, the storyline from MAML to TANML provides little value to me as a reader. The proposed algorithm resembles a classical kernel-weighted linear combination of parameters and the pathological toy experiments provide little value for assessing the actual usefulness of TANML in realistic meta-learning scenarios. However, using the kernel between the task loss gradients as a similarity metrics of task is a nice idea and is a valuable contribution. I highly encourage the authors to further improve the paper. Overall, TANML has scientific merit - when introduced with a convincing storyline and properly supported by realistic experiments and relevant baseline comparisons, this would be a clear accept.

=== Minor remarks ===

- Section 2: Eq. 1: move the comma. It should be $[\theta_0^\top, \nabla_{\theta_0} \mathcal{L}$ …
- Section3: Either the $\Psi$ should be a $T \times D$ matrix, or there should be no transpose in Eq. 2
- Section 3 Eq 2: The kernel in the sum should probably be between i and i’, not between i and i.
- Section 4.1, 2nd paragraph: “... could be ascribed [to] its linear nature …”


[1] Bonilla, Edwin V., Kian M. Chai, and Christopher Williams. "Multi-task Gaussian process prediction." Advances in neural information processing systems. 2008.

[2] Micchelli, Charles A., and Massimiliano Pontil. "Kernels for Multi--task Learning." Advances in neural information processing systems. 2005.

[3] Rothfuss, Jonas, Vincent Fortuin, and Andreas Krause. "PACOH: Bayes-Optimal Meta-Learning with PAC-Guarantees." arXiv preprint arXiv:2002.05551 (2020).

---

> ### Author Response · Authors · 2020-11-25
> **Authors' response to AnonReviewer4**
>
> **Authors**
>
> We would like to thank the reviewer for evaluating our work and sharing their feedback.
>
> **Reviewer**
>
> _Unfortunately, the relationship between Generalized Meta-SGD to TANML is unclear... it is unclear how TANML relates to Meta-SGD.... For instance, if TANML is a generalization of MAML, it would be good to state with which particular choices_ $\theta_0$ and $\pmb\Psi$, _we can recover MAML._
>
> **Authors**
>
> We thank you for the valuable feedback. We have now included a discussion that brings out the explicit relationship between TANML, Meta-SGD, and MAML. Please see the last sentence of the paragraph following Eq (2), and the new Section A of the Appendix.
>
> **Reviewer**
>
> _The related work section is quite minimalistic. For instance, discussing how TANML is different from e.g. multi-task nonparametric methods (e.g. [1-2]) that also use a kernel between tasks, would better clarify how TANML relates to previous work._
>
> **Authors**
>
>  Thank you for the comment. Multi-task nonparametric methods address an entirely different setting $-$ for a given input $x$, the goal is to model the associated vector target $\mathbf{y}$ using kernel regression/Gaussian process. Each component of the vector  target is referred to as a 'task', and multi-task learning thus deals with predicting different variables or components at once using mutual correlation in form of matrix kernels. In contrast, in the meta-learning setting, a task refers to a learning problem by itself with its associated input-output data ( as we have mentioned in Section 1.1). In our approach, the kernel regression adaptation is used to predicts the optimal parameters $\pmb\theta$ for a given task by taking the gradient of the loss function $\nabla\mathcal{L}$ as the input variable to the kernel. Since multi-task learning and our approach address completely different problems even in terms of what they refer to as tasks, we have not included the works on multi-task learning in the related works.
> 	However, we have now included multiple new relevant works under the related works. In particular, we have discussed our approach in the context of a recent related work on kernels in meta-learning. Please see Section 1.2 of the revised manuscript.
>
> **Reviewer**
>
> _The numerical experiments are very simple / limited and designed in a pathological way....
> A real-world use case in which we expect to see a meta-training set with e.g. outliers similar to experiment 2._
>
> **Authors**
>
> Thank you for the comment. We have now included new experiments on real-world regression datasets and on few-shot learning.  Please see the newly added paragraphs on Experiments 3 and 4 in Section 4 of the revised manuscript. We have also added the missing information regarding the numerical experiments in the main text and in the appendix.
>
> **Reviewer**
>
> _Experiments with real world meta-learning datasets. For real-world \& small-scale meta-learning environments for regression, see e.g._[3].
>
> **Authors**
>
> Thank you for the suggestion. We have now included experiments on time-series prediction for the Physionet 2012 Challenge dataset for real-world regression tasks as recommended by you, and on the Omniglot dataset. Please see the newly added paragraphs on Experiments 3 and 4 in Section 4 of the revised manuscript.
>
> **Reviewer**
> _An additional meta-learning setup without outliers / clusters of meta-learning tasks. This way one can assess how the proposed method compares to MAML/Meta-SGD in standard setting_
>
> **Authors**
>
> Thank you for the suggestion. New experiments on real-world datasets for both regression and classification that are now included, and are performed without the presence of outliers or explicit clusters. Please see the newly added paragraphs on Experiments 3 and 4 in Section 4 of the revised manuscript.
>
> **Reviewer**
>
> _Adding missing details, e.g. to the appendix, which are necessary for reproducing the experiment._
>
> **Authors**
>
>  Thank you, we have now added the missing details in the Appendix.
>
> **Reviewer**
>
> _Overall assessment: I vote for rejecting the paper... Overall, TANML has scientific merit - when introduced with a convincing storyline and properly supported by realistic experiments and relevant baseline comparisons, this would be a clear accept._
>
> **Authors**
>
> We thank you for the detailed assessment. We have now included new experiments on real-world data including the Omniglot dataset. We have also discussed the explicit connection of TANML to Meta-SGD and MAML, and included an ablation study of some of the aspects of our approach. Please see the newly added paragraphs on Experiments 3 and 4 in Section 4, and Section A of the Appendix in the revised manuscript.
>
> **Reviewer**
>
> _=== Minor remarks ===_
>
> **Authors**
>
> Thank you for giving it a careful consideration. All the minor remarks have been fixed and we have carefully gone through the manuscript for other typos.

---

### Official Review · AnonReviewer3 · 2020-10-28
**Interesting theoretical formulation, but insufficient experimental analysis**

**Rating:** 4
**Confidence:** 3

**Review:**

This paper proposes a theoretical formulation for meta-learning that uses task similarity based on task gradients, which helps learning in the presence of outlier tasks. The inner loop parameter update is given by linear kernel regression, where the kernel function computes similarity between gradients of different tasks. While the paper includes experiments that outperform MAML and Meta-SGD on estimating randomized linear predictors, and randomized sinusoids with outlier data-points, these are not sufficient to establish the efficacy of the approach.

Pros :
1. This paper proposes a solution to the problem of meta-learning with dissimilar tasks, which is a central problem in meta-learning. The formulated approach is a generalization of MAML and Meta-SGD, as the update direction isn't necessarily the gradient direction, and there is an additional regulation term on meta-parameters in the outer loop. Tasks with similar gradients have a similar update in the meta-learning inner loop.

2. The formulation involving linear kernel regression which enables including the task similarity in the kernel function seems novel and could provide the basis for subsequent work in the field for dealing with dissimilar tasks. The usage of similarity between task gradients to guide updates is similar in spirit to gradient projection techniques used in continual learning to avoid catastrophic forgetting.

3. The included experiments do show much superior performance to MAML and Meta-SGD on the datasets considered, which included tests with outlier data points for sinusoid regression.

Cons :
1. While the experiments show some promise for the method, these on simplistic datasets involving synthetic datasets for estimating randomized linear and sinusoid predictors. Given that the paper discusses MAML and Meta-SGD in some detail for setting up the new method, experiments on the Omniglot and MiniImagenet datasets considered in both those papers would help to better evaluate the proposed approach.

2. The paper has no ablations or analysis for particular parts of their method, such as removing the gradient from the kernel function or removing the regularization term from the outer loop. Thus even on the simplistic datasets considered, it is hard to judge which aspects of the method make it work better.

I am willing to increase my score if the authors include experiments on the datasets mentioned above, and include additional analysis/ablations of their method.

---

> ### Author Response · Authors · 2020-11-25
> **Author's response to AnonReviewer3**
>
>
> **Authors**
>
> We would like to thank the reviewer for evaluating our work and sharing their feedback.
>
> **Reviewer**
>
> _While the experiments show some promise for the method, these on simplistic datasets involving synthetic datasets for estimating randomized linear and sinusoid predictors. Given that the paper discusses MAML and Meta-SGD in some detail for setting up the new method, experiments on the Omniglot and MiniImagenet datasets considered in both those papers would help to better evaluate the proposed approach_
>
> **Authors**
>
> We thank you for the detailed assessment. We have now included new experiments on the Omniglot dataset. Due to limitations on available computational resources, we are yet unable to perform the experiments on the miniImagenet dataset. We have also included new experiments on various real-world regression tasks in the revised manuscript. Please see the newly added paragraphs on Experiments 3 and 4 in Section 4 of the revised manuscript.
>
> **Reviewer**
>
> _The paper has no ablations or analysis for particular parts of their method, such as removing the gradient from the kernel function or removing the regularization term from the outer loop. Thus even on the simplistic datasets considered, it is hard to judge which aspects of the method make it work better._
>
> **Authors**
>
> We thank you for the suggestion. We have now included new section on Ablation study. Please see Section 4  and Section A of the Appendix.
>
> **Reviewer**
>
> _I am willing to increase my score if the authors include experiments on the datasets mentioned above, and include additional analysis/ablations of their method_
>
> **Authors**
>
> We thank you for your valuable feedback. We have now included new experiments on real-world regression tasks, and also on the Omniglot dataset. Please see newly added paragraphs on Experiments 3 and 4 in Section 4 of the revised manuscript. We have also included new sections analyzing our approach and its explicit connections with Meta-SGD and MAML.  Please see Section A of the appendix.

---

### Decision · Program_Chairs · 2021-01-07
**Final Decision**

**Decision:**

Reject

**Comment:**

This paper addresses a method that incorporates the task-similarity (via task gradients) into the meta-learning. The inner loop update is done by kernel regression with the similarity between gradients of tasks considered, and the outer loop is the gradient update with a particular regularization. Without any doubt, it is a timely and important topic to develop a meta-learning method in the presence of outlier tasks. All reviewers criticized the experiments were done on only simple datasets without any ablation study. Authors revised their manuscript, to include more experiments, and tried to clarify the relation of their method to MetaSGD. Unfortunately, however, even after the author response, reviewers were not convinced that their concerns were resolved. In particular, it was claimed that the revised version still lacks comparisons to previous relevant work.